# Structural Plasticity Is a Feature of Rheostat Positions in the Human Na^+^/Taurocholate Cotransporting Polypeptide (NTCP)

**DOI:** 10.3390/ijms23063211

**Published:** 2022-03-16

**Authors:** Melissa J. Ruggiero, Shipra Malhotra, Aron W. Fenton, Liskin Swint-Kruse, John Karanicolas, Bruno Hagenbuch

**Affiliations:** 1Department of Pharmacology, Toxicology and Therapeutics, The University of Kansas Medical Center, Kansas City, KS 66160, USA; melissa.ruggiero@labcorp.com; 2Program in Molecular Therapeutics, Fox Chase Cancer Center, 333 Cottman Avenue, Philadelphia, PA 19111, USA; drshipramalhotra@gmail.com (S.M.); john.karanicolas@fccc.edu (J.K.); 3Department of Biochemistry and Molecular Biology, The University of Kansas Medical Center, Kansas City, KS 66160, USA; afenton@kumc.edu (A.W.F.); lswint-kruse@kumc.edu (L.S.-K.)

**Keywords:** rheostat, transmembrane protein, protein plasticity

## Abstract

In the Na^+^/taurocholate cotransporting polypeptide (NTCP), the clinically relevant S267F polymorphism occurs at a “rheostat position”. That is, amino acid substitutions at this position (“S267X”) lead to a wide range of functional outcomes. This result was particularly striking because molecular models predicted the S267X side chains are buried, and thus, usually expected to be less tolerant of substitutions. To assess whether structural tolerance to buried substitutions is widespread in NTCP, here we used Rosetta to model all 19 potential substitutions at another 13 buried positions. Again, only subtle changes in the calculated stabilities and structures were predicted. Calculations were experimentally validated for 19 variants at codon 271 (“N271X”). Results showed near wildtype expression and rheostatic modulation of substrate transport, implicating N271 as a rheostat position. Notably, each N271X substitution showed a similar effect on the transport of three different substrates and thus did *not* alter substrate specificity. This differs from S267X, which altered *both* transport kinetics and specificity. As both transport and specificity may change during protein evolution, the recognition of such rheostat positions may be important for evolutionary studies. We further propose that the presence of rheostat positions is facilitated by local plasticity within the protein structure. Finally, we note that identifying rheostat positions may advance efforts to predict new biomedically relevant missense variants in NTCP and other membrane transport proteins.

## 1. Introduction

Advances in personalized medicine require computer algorithms that reliably predict which missense variants lead to biomedically relevant changes in protein function. Existing algorithms perform reasonably well for many catastrophic amino acid substitutions (which abolish protein structure or function) and for some neutral substitutions (which have little effect). Substitutions that lead to intermediate functional outcomes are more difficult to identify [1]. Nevertheless, predictions of intermediate functional effects are likely to be particularly important for personalized medicine. While such changes may not cause detrimental effects under “normal” conditions, intermediate effects on function could impact a patient’s therapeutic response to a drug. One such example is the well-known S267F polymorphism of the Na^+^/taurocholate cotransporting polypeptide (NTCP), which has intermediate effects on drug transport [2,3,4].

NTCP is a sodium-dependent bile acid uptake transporter expressed in the basolateral membrane of human hepatocytes. This transporter is important for the enterohepatic circulation of bile acids and also mediates the uptake of certain drugs, in particular the cholesterol-lowering drug rosuvastatin [3,5]. Across the human population, the gene for NTCP has been found to have both variants of medical relevance and variants of unknown significance. For example, two medically relevant polymorphisms in NTCP are S267F and I223T. NTCP S267F, found in up to 7% of Asian subjects but at less than 1% in other populations, cannot efficiently transport bile acids; as a result, patients have elevated levels of bile acids in plasma. The other NTCP polymorphism, I223T, occurs in up to 5.7% of African subjects. This membrane-exposed variant leads to reduced expression on the cell surface and consequently to reduced uptake of bile acids [2]. In addition, more than 300 NTCP missense variants of unknown medical significance are reported in the gnomAD database (Available online: https://gnomad.broadinstitute.org/ accessed on 14 March 2022 [6]). Furthermore, a recent study characterizing rare genetic variants in NTCP concluded that available algorithms were not robust enough to replace functional studies for assessing variant outcomes [7].

With the long-term goal of improving NTCP polymorphic predictions, we previously completed a comprehensive study of all variants at position 267 [8]. The purpose of that study was to assess whether this position has any distinctive properties that could be extrapolated to variants at other NTCP positions. Results showed that, in addition to the phenylalanine polymorphism, most other substitutions at position 267 lead to intermediate outcomes on drug and steroid transport [8]. Indeed, the other 18 amino acid substitutions sampled a wide range of transport abilities, which qualified position 267 as a “rheostat” position [8].

Rheostat positions are a special class of amino acid positions for which varied substitutions can be ordered to reveal a continuum of functional outcomes—ranging from “better” than or similar to wildtype, to partially diminished, to “dead” (inactive) [9]. Interestingly, the substitution rank orders observed for rheostat positions often do not correlate with the expected biochemical classifications of side chains, and other unexpected mutational characteristics such as context dependency have been observed [9,10]. Furthermore, rheostat positions have been identified in numerous proteins and even in multiple positions within the same protein. For example, at least 40% of the amino acid positions in the model bacterial protein LacI have been identified as rheostat positions [11]. Their presences in these proteins indicate that rheostat positions, and the implications of their existence, are widespread not only in numerous proteins but also within a single protein [8,9,12,13,14,15,16].

Thus, we hypothesized that NTCP also likely contains multiple rheostat positions. If correct, we reasoned that systematic substitution studies of these positions would help improve functional predictions for variants of this key transport protein. More specifically, since substitutions at individual rheostat positions “titrate” the affected functional parameter(s), we hypothesized that systematic changes may illuminate the noncanonical, underlying biophysical characteristics that must be understood to improve predictions. Indeed, our studies of NTCP position 267 revealed a striking structural characteristic: even though this position was buried within the protein, structure and energetic modeling showed that NTCP readily adapted to the wide range of side chain chemistries at position 267 [8]. 

As a transport protein, NTCP requires at least two conformations to carry out its function: an outward-open conformation allows for substrate binding outside the cell and an inward-facing conformation allows for substrate release into the cytosol. While crystal structures of human NTCP are not yet available, structures are known for bacterial homologs in the two conformations [17,18]. Thus, in our prior work, we created a homology model of NTCP using the bacterial homologs to assess the structural impacts of substitutions at position 267. Using this model, we determined that position 267 was buried close to the substrate translocation pathway [8]. Mutations that occur in buried locations are normally expected to be highly destabilizing, and thus, are expected to diminish (or abolish) function [19,20]. However, this was emphatically *not* the case for position 267 in NTCP. All substitutions at this position were well tolerated in both the inward- and outward-open conformations. Predicted structures showed localized rearrangements to accommodate changes in amino acid side chains, but as a whole, the protein structures were not globally disrupted. This was confirmed experimentally, as most substitutions at position 267 were well expressed on the cell surface. 

To determine whether this structural plasticity was widespread within NTCP, we performed additional energetic calculations for all 19 potential substitutions at an additional 13 buried positions. All positions were predicted to have a high tolerance for a wide variety of substitutions, including proline. Of these, position 271 was selected for experimental verification of protein expression and transport function. Structurally, position 271 is predicted to be close to 267, buried within the structure, and not facing the predicted substrate binding site/translocation pathway. As reported herein, expression and altered transport were characterized for all 19 NTCP variants at position 271 (“N271X”) and detailed transport kinetic measurements were performed for selected variants. Like position 267, NTCP position 271 acts as a rheostat position for substrate transport. Combined with the structural and energetic modeling, these results suggest that localized structural accommodation of multiple side chain chemistries is a hallmark of, and perhaps a requirement for, the existence of functionally modulating rheostat positions within a protein.

## 2. Results

### 2.1. Structural Modeling and Computed Energetics for NTCP Variants

To determine whether buried positions in NTCP can generally accommodate substitutions without destabilizing the overall protein structure, we computationally substituted 13 additional, buried positions with the other nineteen amino acids, and calculated changes in their energy scores with the Rosetta macromolecular modeling suite. These energy scores, in theory, would correlate to protein stability. As computational controls, the wildtype amino acid was also used as a “substitution” at each position (e.g., N271N and S267S). Results show that the vast majority of variants were well tolerated at the modeled positions (Appendix A). Strikingly, even proline was tolerated, albeit with the most detrimental energy change of the substitutions, at most positions except for position 93. 

Thus, like position 267, the buried positions evaluated here showed localized structural plasticity. This may enable these buried NTCP positions to serve as functional rheostat positions. Alternatively, these positions might tolerate any substitution because they are “neutral”, with their substitutions having no effect on function [21]. These two options must, therefore, be tested experimentally. 

As a first test to determine which of the 13 buried positions are rheostatic or neutral in nature, we chose to experimentally evaluate position 271. In addition to being a buried position, position 271 is structurally close to position 267 (Figure 1) but is predicted to be located away from the substrate binding site. Therefore, unlike position 267, substitutions at this position should not directly interact with translocation of substrates, which make substitution outcomes more difficult to predict. In addition, analyses in our prior work [8] showed that S267 had an evolutionarily ConSurf [22] score of 8 [8] and position 271 has a score of 6. These scores are within the distribution of scores associated with rheostat positions in the E. coli lactose repressor protein and human liver pyruvate kinase, indicating that position 271 was also likely to be a rheostat position. In contrast, functionally neutral positions usually have lower ConSurf scores [11,21].

### 2.2. Protein Expression of N271 Variants on the Cell Surface

To confirm the computed stability, it would be ideal to experimentally measure stability for each of these variants. Such measurements are difficult to directly perform for most integral membrane proteins; nevertheless, a destabilized protein should be more prone to in vivo degradation and, as a consequence, less of the protein would be transported to and detectable at the cell surface. Thus, we utilized the surface expression of NTCP variants to experimentally detect whether substitutions at position 271 grossly destabilized the protein. To that end, we mutated N271 to all other nineteen amino acids, transfected them into HEK293 cells, and determined protein expression at the plasma membrane using surface biotinylation experiments. Figure 2A shows surface-expressed NTCP variants at around 48 kDa; NTCP contains two N-linked glycosylation sites which cause the protein to appear as a “smear” [23]. The Na^+^/K^+^-ATPase α_1_ subunit (sharp band near 100 kDa) was used as a loading control. Quantification of these bands revealed an approximately three-fold difference between the lowest and the highest expressed N271 variants (48% to 160%, with wildtype set at 100%) (Figure 2B), which is comparable to the range of changes previously observed for rheostat position 267 [8]. For position 271, surface expression showed no correlation with predicted stabilities (Appendix A). Nonetheless, these modest experimental changes support the hypothesis that the structural regions around 271 can accommodate a wide range of amino acid substitutions and confirm the small, computed changes in protein stability.

### 2.3. Substrate Transport by N271 Variants

To determine whether position 271 behaves as a rheostat position for modulating transporter function, transport experiments were carried out using three model substrates (Figure 3): taurocholate (top row), estrone-3-sulfate (middle row), and rosuvastatin (bottom row). To isolate changes in transport from the modest changes in surface expression, results from initial uptake experiments (Appendix A) were corrected for surface expression (Figure 2) and the normalized values are shown in Figure 3. As expected for a rheostat position, the variants at position 271 sampled a wide range of the functional outcomes, from approximately 250% of wildtype transporter function down to 40% (excluding proline) (Figure 3). It might be tempting to dismiss the diminished transport of the proline-substituted protein as arising from distortions to the polypeptide backbone that prevent proper NTCP folding. However, note that N271P was glycosylated and trafficked to the membrane at wildtype levels (Figure 2). As such, its structure was unlikely to be globally disrupted and the proline must instead inhibit conformational or dynamic changes needed for transport. 

For the overall set of N271 variants, the amino acid rank orders for the three substrates were highly correlated among the variants (Figure 4 and Appendix A, Appendix A). This indicates that amino acid substitutions at position 271 neither disrupted the substrate binding site nor the translocation pathway for any of the three substrates examined. In other words, changes at position 271 appeared to alter transport but *not* substrate specificity. This result was in contrast to the substrate-dependent changes observed for the S267 variants, which showed both altered transport and rank order and, thus, exhibited altered substrate specificity [8].

### 2.4. Kinetic Characterization of Select N271 Variants

Changes in substrate transport (Figure 3) could arise from changes in (1) substrate affinity, (2) turnover number, or (3) both parameters. To better understand which biochemical parameters were affected by the N271X variants, we selected three representatives for detailed kinetic studies of substrate transport and compared them to wildtype (Figure 5 and Table 1) [8]. First, cysteine was selected because it demonstrated unique changes for all three substrates: similar uptake as wildtype for taurocholate, slightly reduced uptake of estrone-3-sulfate, and the lowest uptake of rosuvastatin (excluding proline). For N271C, kinetic experiments showed that transport capacity (V_max_/K_m_) for estrone-3-sulfate was nearly double that of wildtype and rosuvastatin transport had significant changes in V_max_ and a small change in K_m_ (Figure 5 and Table 1). 

Second, the histidine variant was chosen because its transport was consistently (albeit moderately) diminished relative to wildtype for all three substrates. For N271H, statistically significant increases were detected in V_max_ for estrone-3-sulfate and rosuvastatin but not for taurocholate (Figure 5). Finally, the leucine variant was chosen because it demonstrated increased initial transport for all three substrates. In agreement with the initial transport values, kinetic analyses of N271L showed that the transport capacity was elevated for all three substrates, ranging from ~2-fold for taurocholate to almost 5-fold for estrone-3-sulfate and rosuvastatin (Figure 5).

For all variants, correlation plots for the Michaelis constants versus initial transport rates (Figure 6) suggested that changes to maximal transport velocity (V_max_) were more significant than changes in K_m_. Thus, position 271 may exert most of its effects through altered substrate turnover. This possibility is consistent with the observation that mutations at position 271 alter transport but not substrate specificity.

## 3. Discussion

### 3.1. Local Plasticity May Be a Key Feature of Rheostat Positions

One of the surprising features of rheostat positions has been that they tolerate a wide range of amino acids, modulating function without any evidence that they significantly alter global protein structure. One potential molecular explanation for why this occurs may be that rheostat positions are located in regions with local structural “plasticity”. Several studies, including the one presented here, have generated both indirect and direct evidence for this hypothesis.

Indirect evidence arises from the fact that, at a rheostat position, some main functions remain intact for most substitutions, whereas other aspects of function may be impacted. For example, in ten engineered LacI/GalR transcription repressor proteins, rheostat variants could bind DNA (main function) even if they could not repress transcription. This indicates that their DNA binding domains were folded even if DNA binding affinity was weak [9,24]. Likewise, most rheostat variants at NTCP positions 267 and N271 are effectively post-translationally modified and trafficked to the plasma membrane, regardless of whether their transport was impaired [8]. In human liver pyruvate kinase, variants that altered—and even abolished—allosteric regulation nevertheless retained the main function of enzymatic activity [11,15]. This indicates that the structure of the active site (and presumably the rest of the protein) must be intact. 

Direct evidence for structural plasticity is obtained from comparisons of experimentally determined structures. In comparisons of LacI/GalR paralogs, the linker region that contains many rheostat positions [9] showed only local changes that did not propagate to the overall C^α^ RMSD [25]. For human aldolase A, structures of rheostat variants showed very few changes among their crystallized conformations [26]. 

In an attempt to establish a method to predict which positions would result in mutational tolerance and structural plasticity, we explored whether NTCP rheostat positions could be identified by using structure-based modeling and energy calculations. Indeed, our Rosetta modeling suggested that many buried positions in NTCP tolerated a wide range of amino acids without destabilizing the protein. As with position 267 [8], experimental characterizations of N271 variants demonstrated that this position is a “functional rheostat” position, altering substrate transport. Thus, the results support the emerging conclusion that local structural plasticity is a general feature of protein regions that contain rheostat positions. In future studies, it will be interesting to determine whether the other 12 buried NTCP positions that show structural plasticity are also functional rheostat positions.

Although structural plasticity may be required for a position to act as a functional rheostat position, it cannot be the only defining feature. Structural plasticity must also occur at neutral positions, which accept any amino acid with little effect on function [21]. Thus, the neutral and rheostat positions must have some other underlying biophysical difference besides local structural tolerance to substitutions. One strong candidate is that substitutions at rheostat positions could alter functionally relevant dynamics [26,27]. Since dynamic conformational changes are key to transporter function, transporters—and multi-specific transporters especially—may contain a high density of rheostat positions.

### 3.2. Implications for Understanding Evolution and Advancing Engineering

It is interesting to further speculate whether rheostat positions key for enabling protein “evolvability”, i.e., the “capacity to generate heritable phenotypic variation” [28]. As noted in the Results section, the evolutionary record (as extrapolated from protein sequence alignments) can be used to provide some guidance about the locations of rheostat and neutral positions, with rheostat positions showing modest correlation with the branching of a protein’s phylogenetic tree [21,29,30]. Based on phylogenetic analyses of the solute carrier family 10 (SLC10) protein family, both identified rheostat positions in NTCP support this trend.

The SLC10 protein family may also provide an interesting context in which to consider the differences observed for rheostat positions 271 and 267. For the three substrates assessed, substitutions at position 271 alter only transport, whereas substitutions at position 267 alter both transport and substrate specificity. The two types of functional rheostat positions might have played different evolutionary roles as various member of the SLC10 family evolved alternative specific substrate specificities and transport properties [5,31,32]. It will be interesting to compare the prevalence, location, and structural properties of these types of rheostat positions in SLC10 and other multi-specific drug transporter families, such as the organic anion transporting polypeptides (OATPs, SLCO) [33] and the organic anion and cation transporters (OATs and OCTs, SLC22) [34]. Additional information about the evolution of substrate specificity might be gleaned by comparing rheostat positions among transporters with broad (e.g., the SLC10, SLCO, and SLC22 families) and narrow (e.g., the Na^+^/Ca^2+^ exchanger [35]) substrate specificity. 

Furthermore, knowledge of rheostats positions may provide a means to improve function in protein engineering. Statistically, enhancing function is less likely to occur than diminishing function (there are many ways to damage the structure/function relationship), and therefore, harder to engineer. Improved function due to substitution at rheostat positions has been observed in other proteins [10]. Therefore, evolution may target rheostat positions as a means to enhance protein function, and that same strategy may prove useful in focusing directed evolution/protein engineering projects.

### 3.3. Other Emerging Trends Related to Rheostat Positions

Our studies of NTCP rheostat positions support two additional trends that are emerging from studies of rheostat positions in other proteins. First, substitution outcomes at NTCP positions 267 [8] and 271 were pleiotropic, simultaneously altering two or more biochemical parameters in an uncorrelated manner. For example, when in vivo transport was further quantified by measuring V_max_ and K_m_, substitutions at the same position had different effects on these parameters (Appendix A). The pleiotropy was even more complex for position 267, manifesting as altered substrate specificity [8]. Pleiotropic changes in function were also observed for human liver pyruvate kinase [15]. The fact that changes in these parameters were *not* correlated suggests that substitution effects at rheostat positions have a complex biophysical basis. Second, a subset of variants enhanced function relative to wildtype for both rheostat positions in NTCP. This observation is consistent with observations for rheostat positions in several other proteins [9,10,11,15,26]. 

### 3.4. Estimating the Biological Impact of NTCP Variants

In addition to illuminating features of rheostat positions, the current study also sheds light on the functional relevance of the reported—yet previously uncharacterized—NTCP variants in gnomAD: N271S and N271K. In the current study, the serine variant had diminished transport for all three substrates (Figure 4) but also showed enhanced surface expression (Figure 2B). These two parameters combine to generate an in vivo function that is near wildtype (Appendix A). The lysine variant was only moderately damaging to transport and expression (Figure 2B and Figure 3) and was similar to wildtype in vivo (Appendix A). Thus, both substitutions are likely to be benign, although we note that there is a wide variety of potential in vivo substrates that could behave differently.

Our results can be further expanded to make predictions about other variants at position 271. Most seem to show a trade-off between expression and transport, suggesting that they would not be detrimental. The exceptions are (i) aspartate and glutamate, which have both diminished transport and reduced expression, and (ii) proline, which essentially abolishes transport even though its expression is not altered. As expected from these characteristics, these three variants show the least transport (Appendix A). Therefore, should these three variants be encountered in human patients, they are likely to be deleterious. 

In summary, several buried positions in NTCP were predicted to accommodate substitutions without gross protein destabilization, as is expected for rheostat (and neutral) positions. Experimental characterization of the N271X variants confirmed that position 271 is a rheostat position with modulating effects on substrate transport. However, position 271 does not alter substrate specificity, which is in contrast to the previously characterized rheostat position S267 [8]. These examples highlight a need to separately predict (or measure) both structural and functional impacts of amino acid substitutions in order to assess their biomedical relevance. As we continue to build the database of characterized NTCP variants, we expect to be able to define generalizable rules to eventually predict the outcome of amino acid replacements at less conserved positions.

## 4. Materials and Methods

### 4.1. Computational Models of NTCP Variants

Models of inward- and outward-open human NTCP (NCBI: NP_003040) were constructed as reported previously [8]. Modeling of structures of NTCP variants were also generated using the strategy previously reported [8]. Since the modeling in Rosetta for each position involves a different region of the protein structure, the specific degrees of freedom included in the optimization are different: sampling is focused on the region near the mutation, whereas conformational variations at distant parts of the structure are not sampled as extensively. Thus, the variants for each position can be compared to the wildtype “mutant” at that position (e.g., N271N), but energies from models targeting different positions should not be compared to each other. 

### 4.2. Experimental Materials 

Radiolabeled [^3^H]-taurocholate (6.5 Ci/mmol) was obtained from PerkinElmer (Boston, MA, USA). [^3^H]-Estrone-3-sulfate (50 Ci/mmol) and [^3^H]-rosuvastatin (10 Ci/mmol) were purchased from American Radiolabeled Chemicals (St. Louis, MO, USA). Rosuvastatin (98% pure) was purchased from Cayman Chemicals (Ann Arbor, MI, USA). All other chemicals, including taurocholic acid sodium salt (97% pure) and estrone-3-sulfate sodium salt (containing 35% Tris stabilizer), were obtained Sigma Aldrich (St. Louis, MO, USA).

### 4.3. Site-Directed Mutagenesis and Uptake Experiments

Site-directed mutagenesis at position 271 of the human NTCP open reading frame was performed as described previously, using primers that flanked the N271 codon by approximately twenty nucleotides [8]. All mutational variants were confirmed by sequencing the complete open reading frame. For uptake experiments, we followed the methods described previously for cell culture, transfection, and uptake experiments [8]. For the kinetics measurements, initial linear rates were determined (Appendix A), and uptake was performed within this initial linear portion for all three variants.

### 4.4. Surface Biotinylation and Western Blotting

Surface biotinylation and Western blotting were performed as described before [8]. In brief, NTCP variants were detected with a mouse antibody against the His-tag (Tetra·His Antibody, QIAGEN, Germantown, MD, USA, Cat. No.34670, 1:2000) and as loading control Na^+^/K^+^-ATPase was detected with a mouse antibody against its α subunit (Abcam, Waltham, MA, USA, ab7671, 1:2000). An HRP conjugated goat anti-mouse secondary antibody was used at 1:10,000 (Thermo Fisher Scientific, Waltham, MA, USA, Cat. No. 31430). Chemiluminescent signals were quantified using a LI-COR Odyssey Fc (LI-COR, Lincoln, NE, USA) and their Image Studio Lite Quantification Software. 

### 4.5. Statistical Analyses

Statistical analyses were performed using GraphPad Prism 9 (GraphPad Software Inc., San Diego, CA, USA). Significance was determined using one-way ANOVA followed by Dunnett’s multiple comparisons for surface expression or Tukey’s multiple comparisons for kinetics. Results were considered significantly different at *p* < 0.05.

## Figures and Tables

**Figure 1 ijms-23-03211-f001:**
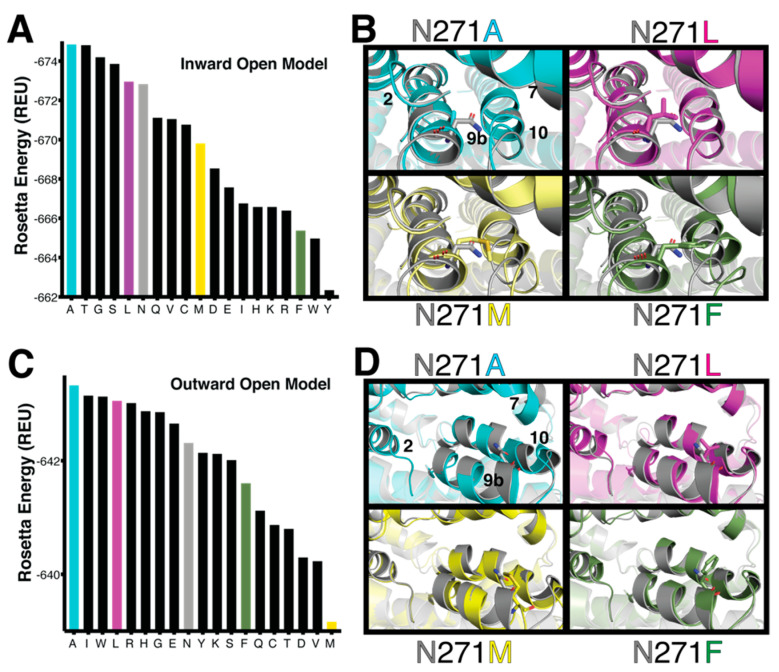
Predicted stability differences associated with sequence variation at the N271 position. (**A**) Rosetta energies for wildtype and 18 sequence variants at position 271 of the inward-open model. Proline is not shown, because incorporation of proline into helical segments cannot be reliably modeled with Rosetta. (**B**) Structural details from the models underlying these energy differences. Four different sequence variants are compared with the wild-type N271; the conformations of TM helices 2, 7, and 10 respond to changes in the amino acid at position 271, which is located on TM helix 9b. (**C**) Rosetta energies using the outward-open model. Proline is again not included in this analysis. (**D**) Structural details from the outward-open models. In this conformation, the positions of TM helices 7 and 10 respond to changes in the amino acid at position 271.

**Figure 2 ijms-23-03211-f002:**
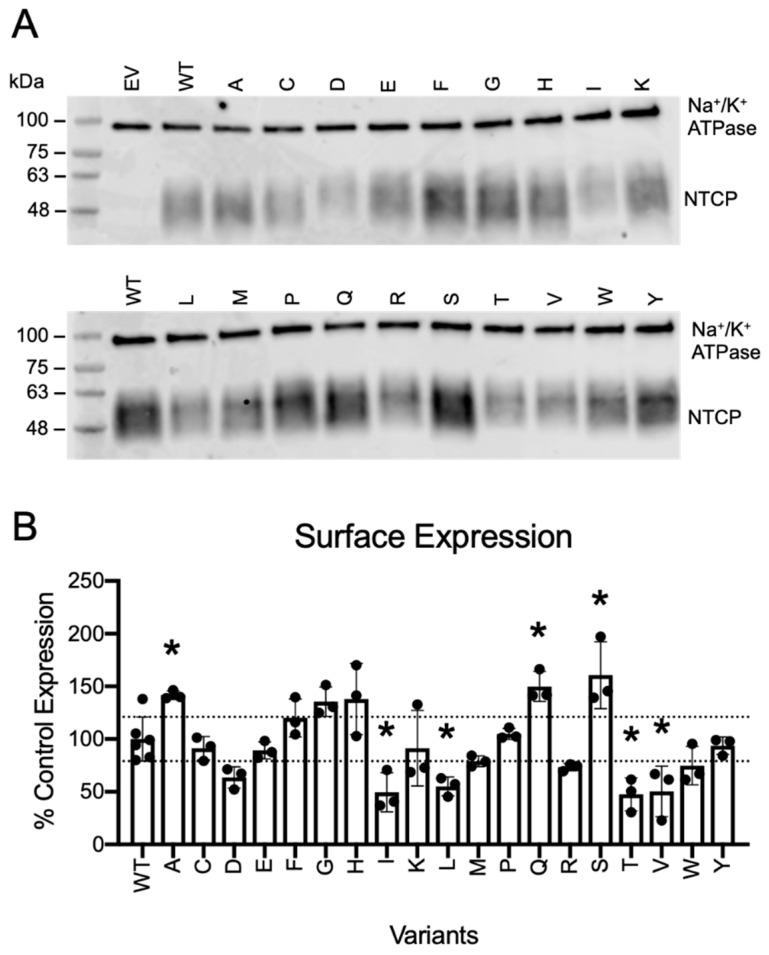
Surface expression and quantification of wildtype NTCP and N271X variants. (**A**) Surface expression of wildtype NTCP and N271X variants on a representative Western blot. Proteins from HEK293 cells transiently transfected with empty vector (EV), wildtype NTCP (WT), and N271 variants were separated using a 4–20% polyacrylamide gradient gel and then transferred to nitrocellulose membranes. Blots were probed simultaneously with Na^+^/K^+^-ATPase as a loading control (100 kDa) and Tetra-His antibodies which detects His-tagged NTCP variants. (**B**) Quantification of Western blots with N271 variants compared to wildtype NTCP. Three independent surface expression experiments were quantified using Image Studio Lite. Individual data points are shown with the bar representing the mean ± SD. Horizontal lines indicate the upper and lower limit of the wildtype value plus (upper line) and minus (lower line) the standard deviation of the wildtype value. Asterisks denote significant difference from wildtype NTCP with a *p* < 0.05 level.

**Figure 3 ijms-23-03211-f003:**
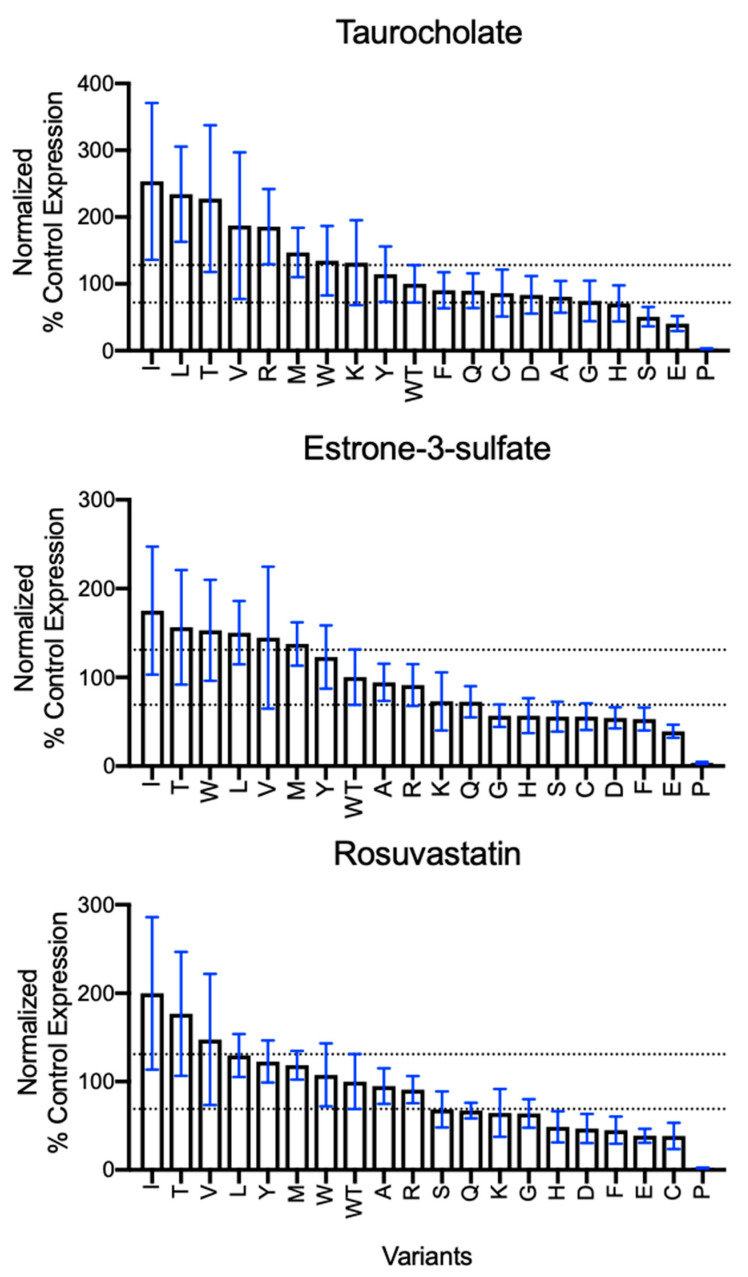
Substrate transport by position 271 variants corrected for surface expression. Initial uptake results from Appendix A were normalized for the surface quantification in Figure 2. Corrected transport is shown with the N271X variants and wildtype NTCP ordered from greatest to least uptake for each substrate. Bar graphs indicate the average of the corrected individual values ± propagated SD. Horizontal lines denote the upper and lower limits of the wildtype standard deviation. Three technical replicates from at least three independent experiments are shown.

**Figure 4 ijms-23-03211-f004:**
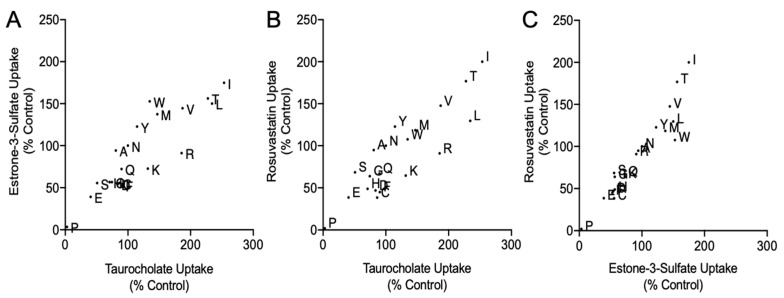
Correlation of normalized variant transport. Surface corrected uptake values from Figure 3 are plotted against each other. Variant amino acid replacements are indicated by their respective letter, with “N” representing wildtype. (**A**) Comparison of ^3^[H]taurocholate to ^3^[H]estrone-3-sulfate transport. (**B**) Comparison of ^3^[H]taurocholate transport to ^3^[H]rosuvastatin uptake. (**C**) Comparison of ^3^[H]estrone-3-sulfate to ^3^[H]rosuvastatin uptake. Linear (Pearson) and rank order (Spearman) correlation calculations were determined and are reported in Appendix A.

**Figure 5 ijms-23-03211-f005:**
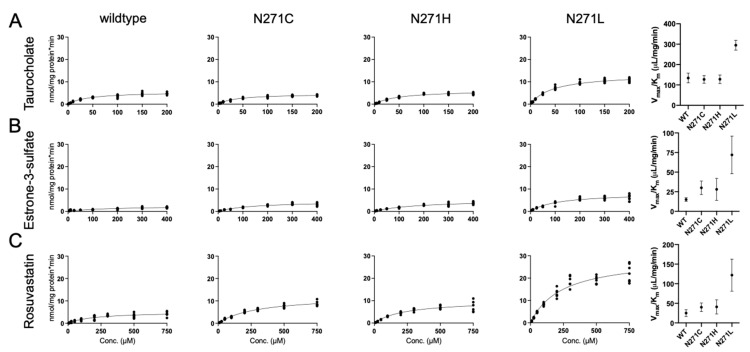
Substrate transport kinetics by wildtype and select 271 variants. Kinetic experiments were measured in HEK293 cells under initial linear rate conditions using increasing concentration of the respective substrates. Kinetics for wildtype (first column) were previously reported [8] and are shown here for visual comparison. NTCP variants N271C (second column), N271H (third column), and N271L (fourth column) were determined using (**A**) taurocholate, (**B**) estrone-3-sulfate, and (**C**) rosuvastatin. The transport capacity (V_max_/K_m_) of all variants and substrates are plotted in the fifth column. The Michaelis–Menten equation in GraphPad Prism 9 was used to determine the curves of best fit, and kinetic parameter results are reported in Table 1. Results were calculated from at least three independent experiments, each with 2–3 technical replicates, and are reported as the mean ± SD.

**Figure 6 ijms-23-03211-f006:**
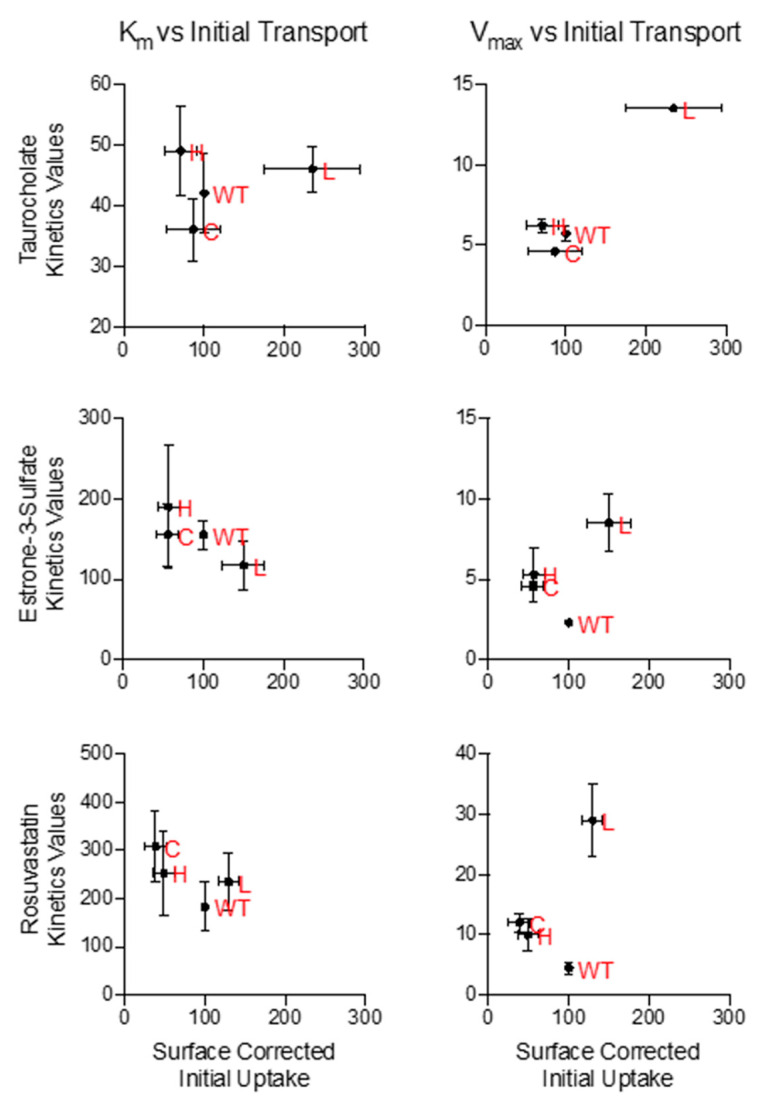
Visual representation of the correlation between kinetic values versus surface corrected initial transport for select N271 variants. Kinetic values, K_m_ (left column) and V_max_ (right column), for select N271 variants from Table 1 were plotted against their surface corrected initial uptake values (Figure 3) for each substrate (taurocholate, first row; estrone-3-sulfate, second row; and rosuvastatin, third row). Comparisons are meant to aid in visual observations of changes in kinetic values. Red letters correspond to wildtype (WT) and the amino acid substitutions in position 271.

**Table 1 ijms-23-03211-t001:** Kinetic parameters for substrate transport by wildtype NTCP and selected variants.

Substrate	NTCP	N271C	N271H	N271L
Taurocholate				
K_m_ (µM)	42 ± 6.6	37 ± 5.1	49 ± 7.3	46 ± 3.7
V_max_ (nmol/mg/min)	5.7 ± 0.5	4.6 ± 0.2 ^#^	6.2 ± 0.4 *	14 ± 0.2 ^#^
V_max_/K_m_ (µL/mg/min)	135 ± 11	129 ± 16	129 ± 13	297 ± 22 ^#^
Estrone-3-sulfate				
K_m_ (µM)	155 ± 18	155 ± 39	190 ± 76	117 ± 30
V_max_ (nmol/mg/min)	2.3 ± 0.2	4.7 ± 0.6	5.3 ± 1.7 *	8.5 ± 1.8 ^#^
V_max_/K_m_ (µL/mg/min)	15 ± 1.0	31 ± 4.8 *	29 ± 2.4 *	72 ± 7.1 ^#^
Rosuvastatin				
K_m_ (µM)	183 ± 51	309 ± 74 *	253 ± 87	236 ± 60
V_max_ (nmol/mg/min)	4.5 ± 1.0	12 ± 1.6	10 ± 2.7	29 ± 5.9 ^#^
V_max_/K_m_ (µL/mg/min)	27 ± 11	41 ± 6.1	42 ± 5.6	126 ± 10 ^#^

Kinetic values, K_m_ and V_max_, calculated using the Michaelis–Menten equation in GraphPad Prism 9 from Figure 5 are reported. Uptake of increasing concentrations of taurocholate, estrone-3-sulfate, and rosuvastatin by HEK293 cells transiently transfected with either empty vector or NTCP variants N271C, N271H, or N271L was measured under initial linear rate conditions, 48 h post-transfection. Net uptake was determined by subtracting transport by empty vector transfected cells from the transport by the N271 variants. Transport capacity (V_max_/K_m_) was calculated by dividing the V_max_ by the K_m_. Kinetic experiments were repeated three individual times with two to three technical replicates completed in each experiment. Results in this table are reported as the mean of those triplicate experiments ± SD. * *p* < 0.05 compared to wildtype NTCP; ^#^
*p* < 0.05 compared to all other variants; Tukey’s multiple comparisons test was used for all statistical analyses. Wildtype NTCP results were previously published [8] and are shown here for comparison.

## Data Availability

PDB coordinates for the two homology models are deposited and freely available on Mendeley online: https://doi.org/10.17632/8bfthm43p5.1 (accessed on 14 March 2022), along with the single lowest-energy Rosetta model for each variant in both the inward-open and outward-open conformations. All other data are contained within the article and in the Appendix A.

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
