# Peer review of "Structural Plasticity Is a Feature of Rheostat Positions in the Human Na+/Taurocholate Cotransporting Polypeptide (NTCP)"

_ijms, 2022, doi:10.3390/ijms23063211_

Round 1
Reviewer 1 Report
Predicting the effect of mutations at conserved position is relatively straightforward and many bioinformatic tools are available for this purpose. Much more difficult is the prediction of the impact of mutations at nonconserved positions. In the last ten years, several papers published protein-engineering works aiming at defining the role of residues present at neutral and nonconserved positions. Indeed, by modifying these residues and observing the effect on the structure or activity of the protein, it is possible to collect data useful in refining algorithms and calculations of the available predictive software. At the end this may help in predicting if some polymorphisms may lead to biomedically-relevant changes in protein function. This manuscript dealing with this topic is the second chapter of a project in which authors intend to construct a database of characterized mutations of the Na+/taurocholate cotransporting polypeptide (NTCP) in order to define generalizable rules to eventually predict the outcome of amino acid replacements at less conserved positions. Authors at first performed an in-silico analysis (using the software Rosetta) of the outcome of introducing all the 19 variations at each one of 13 nonconserved residues. Then they focus on residue N271 and, upon the evaluation of the impact on expression, localization and transport activity of all 19 substitutions, they performed an in-depth characterization of three N271 mutants. They conclude that this position behaves like a rheostat position and presents most of the features of the other rheostat position identified in NTCP (S267) and described in the paper published in JBC in 2020.
Some minor points need to be addressed before publication:
- Lane 170: please refer to table S1 for the correlation coefficients between the computational prediction and the observed differences in stability
- Lane 175: in biotinylation experiments, it would be appropriate to include a control showing the absence of cytoplasmic proteins (i.e. beta-actin) and/or a control of the pull down of non-biotinylated proteins
- Lane 227: provide statistical analysis. Are there any statistically significant differences among samples?
Figure S2: It would be useful for the illustration of the experimental procedure to include in the supplementary data representative time courses of the kinetic experiments of substrates uptake
Author Response
Please see the the attachment.

Reviewer 2 Report
The authors present the results of screening for mutation positions of the NTCP transporter. They have conducted computational mutagenesis trials for the transporter with Rosetta and have identified a number of positions of interest. Furthermore, they have selected one position, for which tey have designed and performed experiments validating their computational predictions.
The manuscript is well written, with minimal English syntax errors that can be easily corrected during proofing. The research design is appropriately presented and results are well supported, both for the computational and the experimental aspects.
I therefore recommend that this manuscript be accepted for publication.
Author Response
We thank reviewer 2 for carefully reviewing the manuscript and we tired our best to correct the minimal English syntax errors.